# What are the modifiable factors of treatment burden and capacity among people with Parkinson's disease and their caregivers: A qualitative study

Qian Yue Tan [1,2]*, Helen C. Roberts[1,2,3], Simon D. S. Fraser[2,4], Khaled Amar[5], Kinda Ibrahim[1,2,4]

1 Academic Geriatric Medicine, Faculty of Medicine, University of Southampton, Southampton, United Kingdom, 2 National Institute for Health and Care Research Applied Research Collaboration Wessex, University of Southampton, Southampton, United Kingdom, 3 National Institute for Health and Care Research Southampton Biomedical Research Centre, University of Southampton and University Hospital Southampton NHS Foundation Trust, Southampton, United Kingdom, 4 School of Primary Care, Population Sciences and Medical Education, Faculty of Medicine, University of Southampton, Southampton, United Kingdom, 5 University Hospitals Dorset NHS Foundation Trust, Bournemouth, United Kingdom

* q.y.tan@soton.ac.uk

**Data Availability Statement:** All relevant data are within the paper.

**Funding:** The study is supported by the National Institute for Health and Care Research (NIHR)

## Abstract

### Background

People with long-term conditions must complete many healthcare tasks such as take medications, attend appointments, and change their lifestyle. This treatment burden and ability to manage it (capacity) is not well-researched in Parkinson's disease.

### Objective

To explore and identify potentially modifiable factors contributing to treatment burden and capacity in people with Parkinson's disease and caregivers.

### Methods

Semi-structured interviews with nine people with Parkinson's disease and eight caregivers recruited from Parkinson's disease clinics in England (ages 59–84 years, duration of Parkinson's disease diagnosis 1–17 years, Hoehn and Yahr (severity of Parkinson's disease) stages 1–4) were conducted. Interviews were recorded and analyzed thematically.

### Results

Four themes of treatment burden with modifiable factors were identified: 1) Challenges with appointments and healthcare access: organizing appointments, seeking help and advice, interactions with healthcare professionals, and caregiver role during appointments; 2) Issues obtaining satisfactory information: sourcing and understanding information, and satisfaction with information provision; 3) Managing medications: getting prescriptions right, organizing polypharmacy, and autonomy to adjust treatments; and 4) Lifestyle changes:

Applied Research Collaboration (ARC) Wessex. QYT received support from the NIHR Applied Research Collaboration ARC Wessex funded through the PhD fellowship. QYT was supported by the University of Southampton NIHR Academic Clinical Fellow training programme. The funders had no role in study design, data collection and analysis, decision to publish, or preparation of the manuscript. The views expressed in this publication are those of the authors and not necessarily those of the National Health Service, the NIHR or the Department of Health and Social Care.

**Competing interests:** The authors have declared that no competing interests exist.

exercise, dietary changes, and financial expenses. Aspects of capacity included access to car and technology, health literacy, financial capacity, physical and mental ability, personal attributes and life circumstances, and support from social networks.

## Conclusions

There are potentially modifiable factors of treatment burden including addressing the frequency of appointments, improving healthcare interactions and continuity of care, improving health literacy and information provision, and reducing polypharmacy. Some changes could be implemented at individual and system levels to reduce treatment burden for people with Parkinson's and their caregivers. Recognition of these by healthcare professionals and adopting a patient-centered approach may improve health outcomes in Parkinson's disease.

## Introduction

Patients with long-term conditions and their caregivers must complete many daily tasks to manage their health. These tasks include taking multiple medications, attending appointments, searching for information, learning about their health condition, changing their dietary intake, and completing recommended exercises [1, 2]. This workload of healthcare and its impact on patient well-being and functioning is termed 'treatment burden' [2, 3]. Eton et al developed a framework of treatment burden which included three themes: 1) work patients must do to care for their health, 2) challenges or stressors that exacerbate perceived burden, and 3) impact of burden [2, 3]. The ability to manage treatment burden ('capacity') can be influenced by multiple factors including physical and mental ability, socioeconomic resources, health literacy and living situation [4–6]. People with high treatment burden or low capacity may have poor healthcare outcomes such as low adherence to treatment recommendations and poor quality of life [1].

Parkinson's disease (PD) is a common neurodegenerative condition worldwide with 6.1 million people living with PD in 2016 [7, 8]. People with Parkinson's (PwP) must manage an array of motor and non-motor symptoms including tremors, rigidity, slowness of movement, sleep disorders, fatigue, urinary and bowel dysfunction, depression, apathy, and psychosis [7, 9]. Management of PD primarily focuses on symptom control, often through multiple medications at different times each day [10]. Additionally, non-pharmacological treatments may be recommended in conjunction with pharmacological treatment in PD [11]. For example, exercise and physical activity supported by physiotherapists and appropriate referral to a multidisciplinary team including an occupational therapist, speech and language therapist or dietician can help manage the progressive symptoms of PD [12]. In a few patients with PD where optimal medical therapy fails to control their symptoms, neurosurgical procedures such as deep brain stimulation (DBS) may be appropriate [13].

Treatment burden and capacity among PwP and their caregivers have been little explored. A recent qualitative systematic review identified potential aspects of treatment burden in PD although none of the included studies specifically aimed to explore the treatment burden or capacity of PwP and caregivers. The review found managing medications, navigating healthcare obstacles at individual provider and system levels, and learning about health were the main aspects of treatment burden [14]. Therefore, this qualitative study aimed to explore the

experiences of treatment burden and capacity among PwP and their caregivers and identify potentially modifiable factors.

## Materials and methods

The study was approved by the National Health Service Research Ethics Committee (21/WM/0058) and is registered on ClinicalTrials.gov (NCT04769973).

### Participant recruitment and sampling

Participants were recruited from two PD specialist outpatient clinics in the South of England, United Kingdom (UK). Inclusion criteria were adults age >18 years with a diagnosis of PD, and/or self-identified caregiver of someone with PD. Exclusion criteria were those who were unable to consent to participate. Purposive sampling was conducted based on age, gender, the severity of PD, and caregiver relationship. Sampling to include patients with PD dementia and caregivers of someone with PD dementia was added after the interviews commenced to capture the potential impact of cognition on treatment burden and capacity experiences. Eligible participants were approached by a researcher (QYT) after their clinic appointment following their agreement with the PD specialist to discuss potential participation in the study. Participants were given a study pack containing a participant information sheet, a reply slip, and a free post envelope. Potential participants were given at least 24 hours to consider their participation and then contacted by QYT (1st author) to answer any questions and arrange an interview date for those interested. PwP were able to participate even if they did not have a caregiver. PwP who had a caregiver were able to participate on their own, and vice-versa. Thirty-two potential participants were invited and 17 (9 PwP and 8 caregivers) consented to participate.

### Data collection

Two interview guides were developed (S1 File): one for the person with PD and one for the caregiver with similar questions on both. The interview guides were developed using Eton's framework of treatment burden, a review of published interview schedules from other qualitative studies of treatment burden and capacity conducted in patients with long-term conditions other than PD, and findings from a systematic review of treatment burden experiences in PD [2, 14–17]. The interview guides were reviewed by our patient and public involvement group (comprising one person with PD, one caregiver of someone with PD, and one caregiver of someone with dementia) which led to additional questions regarding care coordination and changes in question-wording for clarity. The guides were then piloted with two PwP and one caregiver before finalization to ensure ease of understanding and relevance of questions to their experiences of treatment burden and capacity.

One-to-one semi-structured interviews were conducted by QYT at a location and time convenient to participants between July to November 2021. Participants were offered face-to-face, telephone or online video interviews to ensure that data collection could be conducted despite the COVID-19 pandemic. Field notes were taken following each interview to capture initial reflections of treatment burden and capacity aspects within each specific context. The interviews lasted between 45–75 minutes and were audio recorded following written consent. Interviews were then transcribed verbatim by a research assistant and fully anonymized before data analysis.

## Data analysis

Thematic analysis was conducted by QYT and KI (last author) [18]. Nvivo V12 software was used to organize codes and themes. Each transcript was read multiple times, and inductive line-by-line coding was conducted to generate a list of codes and themes alongside the interview field notes and participants' context including length of PD diagnosis, PD severity, and living situation. Discussions between QYT and KI defined and redefined the themes and subthemes to ensure that they reflected the data. This was an iterative process, with multiple diagrammatic mind maps created to visually identify any links and relationships between the themes and subthemes. The findings were further discussed with the research team.

## Reflexivity

QYT is a female medical clinician who conducts regular PD clinics and completed this study as part of a postgraduate degree. None of the participants were known to QYT in a clinical setting before the study, and QYT introduced herself as a researcher at the start of each interview. Prior knowledge of Eton's framework of treatment burden may have influenced data analysis. However, data immersion, maintaining an inductive approach during coding, and multiple discussions between the research team aimed to reduce this potential bias.

## Results

Seventeen participants (9 PwP, 8 caregivers) were interviewed, with 16 interviews conducted face-to-face and one conducted online. Participants were aged 59–84 years, with a duration of diagnosis 1–17 years and H&Y stages 1–4 (Table 1). All participants lived at home; 14 with a

**Table 1. Participants' characteristics.**

| Study ID | Role | Sex | Age (years) | Length of PD diagnosis (years) | H&Y stage | Living situation | Caregiver relationship |
|---|---|---|---|---|---|---|---|
| P01 | Patient | F | 78 | 13 | 2 | Alone | No caregiver |
| P02 | Patient | M | 84 | 3 | 3 | Spouse | Wife |
| P03 | Patient | M | 78 | 1 | 3 | Spouse | Wife |
| P04 | Patient | F | 79 | 10 | 4 | Spouse | Husband |
| P05* | Patient | M | 72 | 17 | 4 | Spouse | Wife |
| P06 | Patient | M | 71 | 4 | 1 | Spouse | Wife |
| P07 | Patient | F | 82 | 5 | 3 | Alone | Daughter |
| P08 | Patient | F | 72 | 11 | 3 | Spouse | No caregiver |
| P09† | Patient | M | 72 | 4 | 3 | Spouse | Wife |
| C01 | Caregiver | F | 78 | 1 | 3 | Spouse | Wife |
| C02 | Caregiver | F | 73 | 9 | 3 | Spouse | Sister |
| C03 | Caregiver | M | 70 | 10 | 4 | Spouse | Husband |
| C04 | Caregiver | F | 70 | 13 | 3 | Spouse | Wife |
| C05** | Caregiver | F | 71 | 17 | 4 | Spouse | Wife |
| C06 | Caregiver | F | 67 | 4 | 1 | Spouse | Wife |
| C07 | Caregiver | F | 59 | 5 | 3 | Alone | Daughter |
| C08†† | Caregiver | F | 73 | 4 | 3 | Spouse | Wife |

Caregiver of PwP; F, Female; M, Male; PD; Parkinson's Disease,

*Deep brain stimulation treatment,

**Caregiver of someone with deep brain stimulation treatment,

†, Diagnosed with PD dementia,

††Caregiver of someone with PD dementia

spouse and three on their own. Two participants with PD did not have a caregiver. Caregivers included spouses, a sister, and a daughter of someone with PD. Four patient-caregiver couples participated in the interviews including one couple with DBS treatment and one couple with a diagnosis of PD dementia. On two occasions, it was impossible to interview the person with PD and their caregiver individually leading to occasional interruptions during the interviews.

## Treatment burden

The main factors contributing to treatment burden identified are summarized in the following four themes: 1) Challenges with appointments and access to healthcare, 2) Issues obtaining satisfactory information regarding PD, 3) Managing prescriptions and medications, and 4) Personal lifestyle changes. The themes and subthemes with supportive quotes from participants are summarized in Table 2.

**Theme 1: Challenges with appointments and access to healthcare.** *Organizing routine healthcare appointments*. PwP and caregivers reported attending multiple appointments with various healthcare professionals for their PD such as a PD specialist doctor, PD nurse specialist, General Practitioner (GP), physiotherapist, occupational therapist, psychologist, speech and language therapist, and the older people's mental health team. Negotiating the system for arranging appointments and unexpected changes to planned appointments were reported as challenging for some participants and caused stress and frustration. Participants also described dissatisfaction with the frequency of PD appointments. A few participants preferred more frequent appointments with the PD team, whilst others felt that they had too many appointments which consequently had a negative impact on their personal and social activities. During the COVID-19 pandemic, PwP and caregivers reported cancelled or delayed appointments and telephone appointments rather than face-to-face appointments which contributed to treatment burden. The impact of PD on speech meant that PwP reported that their voices were not heard clearly over the telephone. Furthermore, they found it difficult to describe their PD symptoms and concerns and felt that they were unable to build rapport with healthcare professionals over the telephone.

*Seeking help and advice from healthcare professionals.* Most participants were able to get in touch with their PD specialist, PD nurse specialist, GP, or pharmacist when they had a concern about their PD. However, some participants reported difficulties accessing their GP due to the healthcare pressures and ten-minute appointment time slots which were insufficient to address their complex health needs with PD. The online GP electronic consultation service was difficult to use due to the slowness of movement and tremors in PD affecting the use of computers. Other participants reported hesitancy in seeking medical advice and chose not to seek help between their planned routine appointments with the PD team unless necessary. Reasons reported were not wanting to bother healthcare professionals, avoiding further tests or disruptions to their daily activities, trying to manage issues on their own by searching for information on the internet, and poor relationships with their GP.

*Interactions with healthcare professionals.* Most participants described that they were able to build trust and relationships with the PD team and that their concerns were listened to and addressed appropriately. Yet, a few participants reported that the lack of care continuity and lack of shared decision-making with healthcare personnel prevented this relationship building. Additionally, a few PwP and caregivers described poor relationships with their GP due to poor communication and continuity of care, a lack of empathy, and a lack of understanding about their health issues with PD.

*Caregiver role during appointments and access to healthcare professionals.* Caregivers played a key role in organizing and attending appointments including helping the person with PD

**Table 2. Examples of participants' quotes of treatment burden in Parkinson's disease.**

| Themes | Subthemes | Issues of Treatment Burden | Supportive Quotes |
|---|---|---|---|
| **Theme 1: Challenges with appointments and access to healthcare professionals** | Organizing routine healthcare appointments | Attending multiple healthcare appointments<br>Negative impact of COVID-19 on quality and frequency of appointments | *"As far as I'm concerned, it's just one more bl\*\*dy visit to medics of some sort. You know, by the time you've gone to the dentist, opticians, consultants for my eyes, and I've got to go and see the doctor about this, it's probably skin cancer. It's nearly always something on that means I have to go out and spend time doing stuff when I might just like to finish reading my book from the library."* P08<br>*"I still think I'd prefer face-to-face, cos I think body language is a big sign about things. And you can get a better rapport with somebody you're sitting with, rather than this digital."* P06 |
| | Seeking help and advice from healthcare professionals | Methods of contacting healthcare professionals<br>Hesitancy in seeking medical advice<br>Difficulty in accessing GPs for advice | *"And, because I'm very slow and I keep on hitting the wrong button because I'm shaking, so, it's a nuisance to use it (online electronic consultation)."* P08<br>*"Most of it I leave, and I think it will be alright tomorrow sort of thing. But then, if it's quite a while (daughter) says, 'You've gotta do such and such a thing'. She might say, 'Get in touch with the doctor or the nurse or something. Find out what is happening.'"* P07<br>*"It took me a couple of days to get through to them (GP) because they have a different system. If you want such and such press this and if you want, then you're on the phone and waiting and waiting and waiting. I'm waiting my life away, you know."* C02 |
| | Interactions with healthcare professionals | Care coordination between healthcare services<br>Continuity of care and building relationships with healthcare professionals | *"Anything they (PD specialist) send to my GP they seem to ignore. I don't think they even read the letters. I get no reaction from GP at all."* P05<br>*"I used to have a GP, she retired about 3 years ago. And when she was my GP, which she was for about 20 years, she got to know me, I got to know her, and she was a person I went to see. Nowadays, no one doctor knows me. I don't like that."* P08 |
| | Caregiver role during appointments and access to healthcare professionals | Help communicate and raise issues with healthcare professionals<br>Reminding PwP of the outcomes from healthcare appointments<br>Contacting healthcare professionals on behalf of PwP | *"And I've generally gone along with (husband) to his consultation meetings. And, you know, (PD specialist) was very good because she allowed me at times to talk."* C04<br>*"So, I sat in on her (physiotherapy) sessions because mum, unfortunately, is forgetting things now, so I can remind her, yes."* C07<br>*"I was thinking about setting in a care plan and having to deal with doctors, that was the first thing, dealing with the doctors and the medication. And then the council, the frailty team, the nurses that were dealing with him."* C02 |
| **Theme 2: Issues obtaining satisfactory information regarding PD** | Sources of information | Receiving and signposting to information<br>Searching for information<br>Learning from personal and other people's experiences | *"Consultant said to me, 'if you want my advice, learn as much as you can about PD. Read everything you can, try and find the association and learn everything you can so you can make informed choices about your treatment and medication and things like that'. So, I followed his advice."* P05<br>*"And the other stuff, I just learn on the hoof, because we asked (PD specialist) what to expect and the bottom line is that no one person is the same with PD so he couldn't tell us exactly what to expect. So, he wasn't going to frighten us with stuff that could happen but might not happen. So, I think that was the best way round."* C03 |
| | Understanding information and satisfaction with levels of information provided | Understanding information provided<br>Poor levels of information provided<br>Personal preference for information related to PD | *"I have a fair idea about what might happen to me Parkinson's wise so I can generally tell whether something is or isn't. And if I'm not sure, I don't bother to know, I get on with my life."* P08<br>*"They didn't say, well this is going to happen; that might happen this, they didn't do any of that. They just said, 'yes (husband) you've got Parkinson's, thank you very much'."* C06<br>*"So, the information is out there, it's whether you want it or not. I know several people who don't want to know, whereas I did want to know, and I still want to know."* P01 |

*(Continued)*

**Table 2.** (Continued)

| Themes | Subthemes | Issues of Treatment Burden | Supportive Quotes |
|---|---|---|---|
| **Theme 3: Managing prescriptions and medications** | Getting prescriptions right | Errors in prescriptions<br>Collecting prescriptions | *"I've probably taken 20 minutes running between the pharmacy and the GP, where the GP said something, well via the receptionist, cos you can never see the GP. And then you go back to the pharmacy, and they say 'right, right medication, the right prescription should have come through now', cos they've sent the wrong prescription for whatever reason. And um, you get to the pharmacy, and they say, 'it's not come through, it must be in the ether somewhere'." C07* |
| | Managing polypharmacy and its impact on PwP and caregivers | Taking multiple medications at different times<br>Approaches to help medication taking<br>Monitoring response to treatment and impact of missed medications | *"I'm managing fine except it takes me at least half an hour in the morning to put them together cos I have I think it's about 19 tablets. And it's then that I think, get the tablets container. It says to take one three times a day say, so I get three out, put them in some things where they've gotta go, read it up make sure it's the right one, and I'm over checking myself all the time." P07*<br>*"Suddenly she gets up from the chair and finds she can't walk to the door cos everything's stopped. You know, and that's just the effect of, so yes it does make a difference. Yes, we have been late, but that's when she's really late taking (medications)." C07* |
| | Autonomy to adjust treatments | Seeking advice from healthcare professionals<br>Taking control of PD treatments | *"He started by three a day, and then it went up to four, and then when we saw (PD specialist) last year, he said if he can tolerate having another one twice a day, do it so it's like he's having six a day now." C08*<br>*"I very cheekily altered the (medication) times with what, I don't know who it was, did it for me because they didn't suit me, so I altered them." P01* |
| **Theme 4: Personal lifestyle changes** | Exercising and keeping active | Attending physiotherapy and exercise classes<br>Maintaining physical activity | *"I did go through all the exercises and that with the nurses up there, and I did them quite well. But now, most of the time I'm too weak to do them. Like if I feel weak and I can't be doing it, when I'm feeling better, I want to catch up on something I can do." P07*<br>*"I don't want to eat and put on a lot of weight because that wouldn't be good. That's why I like walking to keep as active as I can." P02* |
| | Dietary changes | Maintaining healthy diet<br>Changes in diet due to PD medications and symptoms | *"I try and be careful what I'm eating. Certain things I try and avoid it if I'm doing anything that requires going out as it interferes with the absorption of Ropinirole. Like cheese, I love cheese, but it blocks the Ropinirole. So that's out now." P05*<br>*"I don't drink, I never go to the pubs or anything. I'm on these pills, why mix it with alcohol? I'm taking pills for a purpose, why interfere with that." P02* |
| | Financial expenses related to health | Expenses for travel to appointments, equipment, mobility aids, lifestyle changes, and practical support for daily activities | *"We've got a bigger shower now. A walk-in shower and aids for (husband). So, we had to have the fourth bedroom smaller to make a really big bathroom for him." C05* |

C; Quote from caregiver, GP; General Practitioner, P; Quote from person with PD; Parkinson's Disease, PwP; People with Parkinson's

communicate with healthcare professionals due to speech difficulties in PD, prompting them to discuss symptoms and medication issues, raising additional issues they have noticed, and reminding the person with PD who had memory issues of the outcomes or management changes following appointments. They also reported accessing and contacting healthcare professionals and city councils on behalf of the person with PD to discuss medication issues or arrange a suitable care plan.

**Theme 2: Issues obtaining satisfactory information regarding PD.** *Sources of information.* Participants reported receiving information from multiple sources following the diagnosis of PD including healthcare professionals, Parkinson's UK (a Parkinson's research and support national charity), family members, or searching for information themselves on the internet. A few participants reported that they were encouraged by their PD specialist doctors and nurses to learn as much as possible about PD such as learning about PD symptoms, medications, and the impact on driving and insurance. Many participants described learning how to manage their health with PD from personal experiences and talking to other PwP and caregivers by attending Parkinson's local support groups as they felt that healthcare professionals

were unable to tell them what to expect. However, a few participants reported that seeing others with more advanced stages of PD reminded them of the potential future deterioration that could happen to them. Some participants reported wanting to know as much information as possible as it helped ease their concerns and manage their PD. Others preferred not to know more than necessary to avoid worrying about the future. It appears their personal preferences about information levels may change over time. For example, one participant with PD searched for much information regarding PD after her initial diagnosis and reported that after 11 years of living with PD she now feels that she has enough information and prefers not to search further.

*Understanding information and satisfaction with levels of information provided.* Some participants were unhappy with the level of information provision, particularly about the potential symptoms of PD, possible worst-case scenarios, prognosis, and long-term future with PD. Caregivers also described poor information on how to care for someone with PD. Out of desperation and uncertainty, some PwP and caregivers searched for information themselves on ways to manage the symptoms of PD, medication side-effects or devices to help with medication adherence using the internet or by going to the library even though they did not particularly want to. PwP and caregivers also described that the information provided could be confusing and difficult to understand, particularly the medical terms used. Poor explanation from healthcare professionals about the diagnosis and possible causes of PD as well as the potential prognosis led to PwP and caregivers feeling unsupported and unable to manage their health with PD.

**Theme 3: Managing prescriptions and medications.** *Getting prescriptions right.* Participants experienced errors in medication prescriptions due to miscommunications between the PD specialist, GP, and pharmacy which were difficult to resolve. A few participants also reported delays in obtaining prescription changes and delays in getting prescriptions ready, resulting in occasions when not all medications were dispensed which may be detrimental to the management of PD. Some PwP were able to order their prescriptions online and collect their prescriptions from the pharmacy. However, other PwP relied on their caregivers or friends to complete this task as they were unable to use a computer themselves due to tremors, had poor memory, and experienced mobility issues due to PD.

*Managing polypharmacy and its impact on PwP and caregivers.* Taking multiple medications at different times each day to manage PD and other long-term conditions such as hypertension, diabetes, hypercholesterolemia, and asthma were challenging for some participants. A few participants also reported the negative impact of medications for other health conditions on their PD, such as experiencing dizziness and low blood pressure exacerbated by medications to treat hypertension. Due to the polypharmacy, PwP and caregivers reported that they had to be vigilant when reading medication names and instructions on the prescriptions to avoid any errors or confusion. Organizing medications was described as time-consuming, and one participant with PD reported that it took up to 30 minutes to do so daily. Some participants accepted that despite the tiresome work of taking many medications, the noticeable positive response of PD symptoms meant they realized that taking PD medications was a necessity and made sure not to miss any doses. Others reported persisting with PD medications despite a lack of improvement in PD symptoms and not noticing any difference with missed or delayed medications. However, the strict adherence to the multiple PD medications timings throughout the day was reported as preventing PwP and caregivers from doing their usual activities. Some PwP also described difficulties managing the varying effects of PD medications on a day-to-day basis as well as experiencing medication side-effects such as hallucinations, depression, irritability, and anxiety.

Participants described approaches to managing polypharmacy such as routinising medication-taking into their daily activities, writing down medication schedules, and using different pill devices and technology such as alarms on an iPad® or reminders on the Alexa® device. Nevertheless, difficulties with fine movements and memory issues due to PD meant that some PwP reported being unable to manage medications on their own. Consequently, caregivers described managing medications by removing medications from packaging, laying medications out during mealtimes, and reminding them of medication times. Moreover, issues with swallowing as PD progressed meant that a few participants described learning new ways of managing medications such as dissolving PD medications in water and using a straw.

*Autonomy to adjust treatments*. Treatment changes could be challenging for PwP and caregivers. One participant with DBS reported multiple adjustments of the DBS device voltage settings, requiring regular six-weekly appointments to achieve adequate PD symptom control. Some PwP reported that they always sought advice from their PD team for any adjustments in PD medications doses and timings but were given final autonomy to make any decisions after considering the benefits and side-effects of medication changes. In contrast, other PwP took control of their medications and changed their medication timings to fit around their planned personal activities despite the instructions on prescriptions. One participant with PD and caregiver discussed any medication changes between themselves and weighed up the potential impact of medication changes.

**Theme 4: Personal life adaptations.** *Exercising and keeping active*. Most PwP and caregivers described trying to keep physically active by walking or gardening as recommended by healthcare professionals. Some participants reported that they were referred for physiotherapy and were given exercises to help improve balance, walking, ability to stand from a chair and appropriate use of mobility equipment aids. However, some PwP did not notice any difference in their symptoms and were unsure if exercise was helpful. A few PwP with mid-to-late stages of PD stated that they were unable to complete the exercises due to symptoms of fatigue and weakness and chose to prioritize other activities when they felt able to. Due to the COVID-19 pandemic, a few participants noticed a deterioration in the mobility of the person PD as they were not able to be as physically active due to the closure of leisure centres, lack of exercise classes, and not going out for walks due to concerns about contracting COVID-19.

*Dietary changes*. Some participants described maintaining a healthy diet with fresh fruit and vegetables and ensuring a stable weight. Other PwP reported avoiding certain food and drinks such as alcohol or cheese by choice, as they found from experience that it exacerbated PD symptoms such as tremors and interfered with PD medications which consequently affected their mobility and ability to carry out daily activities. Swallowing and dexterity issues meant that a few participants described a change to softer meals, and a need for caregivers to cut the food into smaller pieces for the person with PD.

*Financial expenses related to health*. Due to the progression and impact of PD symptoms on mobility, some PwP and caregivers described the financial expenditure for equipment and mobility aids such as a shower stool, shower rails, walker, trolley, or wheelchair to help their mobility, maintain independence and allow them to leave the house for activities. Difficulties completing activities of daily living due to PD symptoms also meant that some participants reported paying for private carers, a cleaner, a gardener, or the delivery of meals to help them manage this. A few participants reported personal home renovations to increase accessibility for the person with PD, adding to the costs of managing their health with PD.

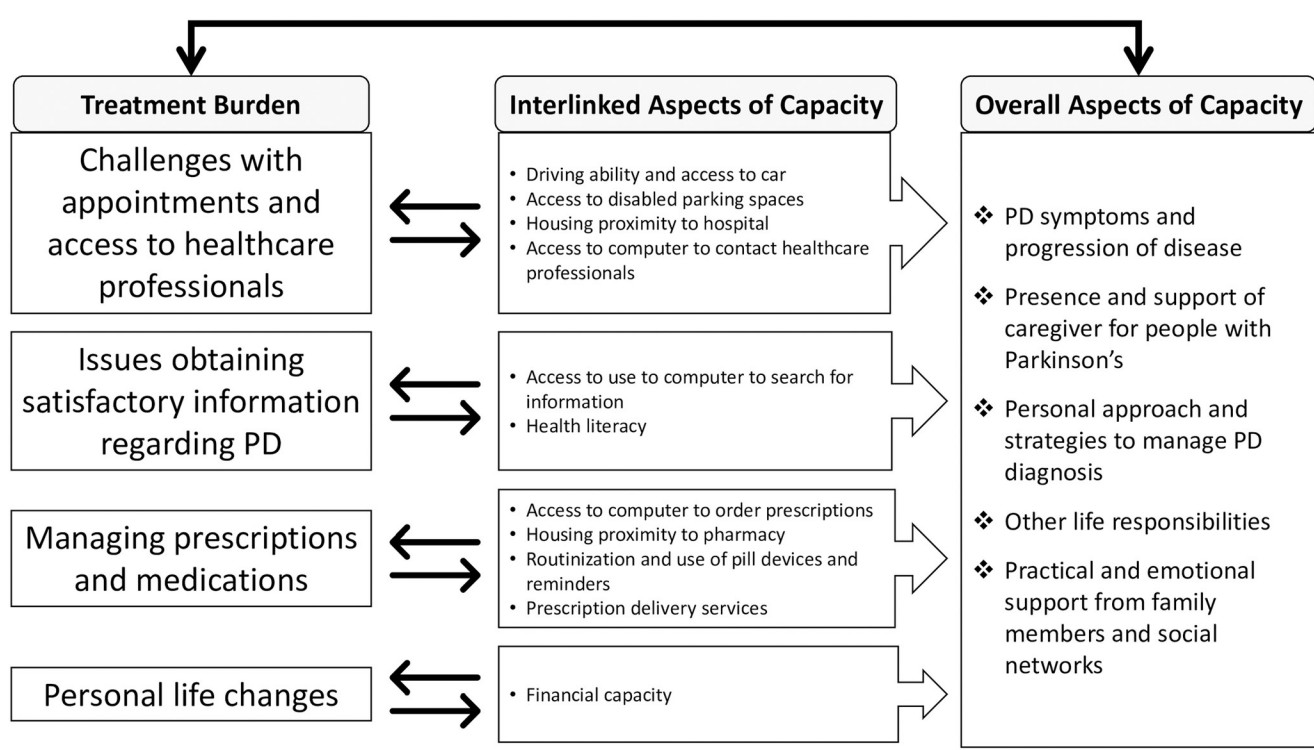

**Fig 1. Interlinked and overall aspects of treatment burden and capacity in Parkinson's disease.**

## Capacity

There were aspects of capacity for PwP and caregivers that specifically related to the issues of treatment burden in PD described in the four themes above as well as the overall capacity of PwP and caregivers (Fig 1). For example, aspects of capacity that enabled PwP and caregivers to get to hospitals for their healthcare appointments were their ability to drive, access a car, and access to disabled parking spaces. The proximity of their homes to the hospital and pharmacy and ease of public transport also helped them access these services. Searching for information and understanding the information related to PD were supported by their health literacy levels, family members, personal life circumstances and experiences. Access and ability to use a computer enabled PwP and caregivers to get in touch with healthcare professionals, search for information and order prescriptions from the pharmacy. The management of prescriptions and polypharmacy for PwP and caregivers were supported by routinization of medication taking into everyday activities, use of pill devices or reminders and prescription delivery services. Furthermore, having greater financial resources enhanced participants' capacity to complete personal life adaptations such as attending private exercise classes, maintaining dietary changes, purchasing equipment or mobility aids and obtaining additional practical support for activities of daily living.

Additionally, other aspects contributed to the overall capacity of PwP and caregivers to manage the treatment burden (Table 3). Firstly, the **presence and support from caregivers** was an important aspect of capacity for someone with PD. Caregivers assisted PwP with accessing healthcare services, managing medications and prescriptions, and helping with understanding of the information provided. Both PwP and caregivers reported the importance of maintaining a strong relationship with each other by ensuring honesty, open communications and working together. Secondly, many participants described their **personal attributes**

**Table 3. Examples of participants' quotes of overall aspects of capacity in Parkinson's disease.**

| Overall Aspects of Capacity | Supportive Quotes |
|---|---|
| **Presence and support from caregivers** | *"I rely upon my family. My memory for going back a long way before I retired and things like that, um, is still quite good. Now, if I'm talking to you sort of thing, I forget where I get to. As I say, it's infuriating the polite way of putting it. But I rely upon my family."* P09<br><br>*"Well, he will take the wrong tablet at the wrong time, um, and you say to him, 'you know, what happened there?'. I mean, even now, with his timer box he's got. Sometimes it'll go off and he'll go out to the kitchen, um, and I think he gets a drink and gets lost and takes a drink and doesn't take tablets. So, I'm always having to look in the little box to check."* C08 |
| **Personal attributes and life circumstances** | *"I think the good thing was I accepted it from the beginning cos I knew there was something wrong and, with the Lord's help I was able to knuckle to and sort myself out and make the most of it."* P04<br>*"I suppose I have no alternative. You have to get on with it; you have to try and manage it the best you can. I think fatigue's the hardest thing, because if you're fatigued you can't do anything, or you feel you can't do anything. Sometimes you have to push yourself."* P06 |
| **Practical and emotional support from family members and social networks** | *"I have surrounded myself with help so, although once COVID came she stopped doing my hair and I found out how to do it myself. So, I did have a hairdresser; I have a gardener; I have a cleaner; I have a window cleaner."* P01<br>*"We (neighbors) all know each other so if you get into a fix like when (wife) fell in the garden and I had to go and get help to try and get her up, and our church home group is very good as well."* C03 |

C; Quote from caregiver, P; Quote from person with PD; Parkinson's Disease

**and life circumstances** that affected their ability to manage PD. Maintaining a positive attitude, a strong sense of independence, a sense of humor, being level-headed and taking each day as it comes helped PwP and caregivers accept the diagnosis, progression, and impact of PD. A few participants also reported the importance of faith and religion in helping them accept the challenges of PD and making the most of their lives. Some PwP and caregivers reported that other life responsibilities such as work, household maintenance and caring responsibilities for elderly parents or grandchildren could impact their ability to manage the treatment burden. Finally, most participants reported the invaluable practical, emotional, and psychological **support from family members and wider social networks** such as friends, neighbors, church members, and local Parkinson's UK support groups. Sharing experiences with other people in the same situation, support getting to exercise classes and hospital appointments along with help with activities of daily living such as washing, dressing, cooking, and gardening not only helped PwP who lived alone, but also other PwP and caregivers manage the healthcare tasks in PD.

## Discussion

This qualitative study has for the first time explored the experiences of treatment burden and capacity in PD. High treatment burden among PwP and caregivers related to challenges organizing and attending multiple appointments, poor access, and interactions with healthcare professionals, difficulties obtaining satisfactory levels of information related to PD, managing prescriptions and medications, and enacting personal life adaptations. Aspects of capacity for PwP and caregivers included driving ability, access to car and technology, living proximity to

amenities, health literacy, financial capacity, personal attributes, and availability of support from family members and social networks. The symptoms and progression of PD such as tremors, poor dexterity, swallowing problems, fatigue and poor memory were reported to impact their ability to manage medications, access healthcare services and complete recommended exercises. This may result in increased treatment burden or reduced capacity in PwP and caregivers. Indeed, treatment burden and capacity appear to be closely interlinked in PD as seen in Fig 1. This aligns with the Cumulative Complexity Model and Burden of Treatment Theory which describes the dynamic relationship and interaction between patient workload and capacity, and the important interactions of social networks including healthcare professionals with this structural model [4, 19].

This study has identified the potentially modifiable factors that could reduce treatment burden or enhance capacity in PD. For instance, rather than offering follow-up appointments for PD at routine intervals, a move towards patient-initiated follow-up appointments where patients or caregivers have control over their follow-up care as recommended by National Health Service (NHS) England in 2020 could address the dissatisfaction with frequency of appointments voiced by PwP and caregivers [20]. However, whilst the benefits of patient-initiated follow-up appointments have been shown in other health conditions such as breast cancer, inflammatory bowel disease and rheumatoid arthritis, it's use in PD remains uncertain [21]. Furthermore, poor interactions and relationships between PD service users and healthcare professionals may be improved through specific training strategies that enhance communication and interpersonal skills of healthcare professionals. This could potentially ensure better patient-centered communication and reduce treatment burden [22, 23]. Aligning appointments, improving access, care coordination, and continuity of care between primary and secondary healthcare services through development and implementation of integrated care models for PD may also improve the treatment burden experiences for PwP and caregivers [24, 25]. Furthermore, tailored information provision by healthcare professionals based on personal preferences and stages of PD and structured medication reviews either by GP, PD specialist or pharmacists to reduce polypharmacy and frequency of medication timings could also be beneficial [26, 27].

Likewise, patient capacity may be enhanced such as improving health literacy through the appropriate provision of information in various modes that are easily accessible to PwP and caregivers, structured education sessions for PwP and caregivers and effective communication from healthcare professionals [26, 28, 29]. Encouraging self-management and change in personal approaches to PD by healthcare professionals or capacity coaching could help PwP and caregivers draw on existing sources of capacity or cultivate new strategies of managing a long-term condition such as PD [30, 31]. Healthcare professionals could signpost PwP and caregivers to information regarding medication aids as a simple way to increase utilization of practical strategies such as pill devices and prescription delivery services to help medication burden. These recommendations for change at individual provider and system levels could improve the treatment burden experiences for PwP and caregivers. However, further research is required to determine the effectiveness of the proposed changes in PD.

Our interview findings align and add to the systematic review that reported the main contributors to treatment burden in PD relate to medications, healthcare obstacles at individual and system levels and information provision [14]. Additional factors relating to prescription errors, medication availability, collecting prescriptions, and issues with access to GP are reported in our study. These issues were also reported in UK studies of treatment burden in patients with other long-term conditions, including stroke and chronic kidney disease [15, 17]. Furthermore, difficulties understanding information were reported by PwP and caregivers, with factors such as previous occupation and family support affecting their ability to

understand health-related information. Awareness of PwP and caregivers' personal preferences for information can help healthcare professionals ensure information provision and explanation at the appropriate level. Studies in patients with multiple long-term conditions in the UK and multimorbid patients with cardiovascular disease in Denmark have found that low health literacy was associated with high treatment burden levels [32, 33]. Therefore, health literacy may be an important and potentially modifiable aspect of capacity as highlighted in our findings [4].

The COVID-19 pandemic may have had an impact on the treatment burden in PD due to necessary changes in healthcare delivery leading to delayed or cancelled appointments and poor experiences with telephone appointments reported by PwP and caregivers. This is in line with findings from a large Parkinson's UK national survey that reported that 34% of respondents had appointments with the PD specialist or PD nurse specialist cancelled during the pandemic, whilst more than half were not offered a telephone or online appointment [34]. Negative experiences with telephone appointments were also reported in other studies with PwP [35, 36]. However, a recent implementation study conducted in Canada reported that the use of virtual visits in primary care may reduce treatment burden related to medical appointments and monitoring health status [37]. Therefore, the use of telemedicine as an adjunct or additional service for clinicians may be beneficial to some PwP with severe disability, homebound or those living in rural areas who have access to internet-enabled devices through reduced travel time and costs [36, 38].

Caregivers and social networks have an important role in supporting someone with PD. The presence of a caregiver was a fundamental aspect of patient capacity and managing treatment burden for the PwP in this study. This is increasingly important due to the progressive PD symptoms. Caregivers themselves experienced treatment burden by attending appointments, managing medications, learning about PD, and enacting lifestyle changes together with the person with PD they support. Caregivers managed this treatment burden on top of providing physical, social, and emotional support, as well as assisting with personal care and activities of daily living [39, 40]. Moreover, some caregivers of PwP may themselves be diagnosed with a long-term condition and have to manage their own health [41]. This can be demanding and contribute to caregiver burden, a well-researched yet separate concept defined as "the extent to which caregivers perceive that caregiving has had an adverse effect on their emotional, social, financial, physical and spiritual functioning" [42, 43]. Although our findings have highlighted caregivers' invaluable role in managing the health of someone with PD, treatment burden amongst caregivers of people with long-term conditions remains understudied [44]. A systematic review of qualitative studies reported that caregivers of patients with chronic obstructive pulmonary disease (COPD) experienced increasing accumulation of treatment burden as the disease progressed with functional deterioration of the person with COPD [45]. This aligns with our study findings that report the impact of increasing symptoms and inevitable progression of PD on the treatment burden and capacity in both PwP and caregivers. In particular, the presence of cognitive impairment and dementia may mean that the person with PD may no longer be able to manage the treatment burden themselves, relying instead on their caregiver to complete the workload of health [46].

## Strength and limitations

A strength of this study is the use of purposive sampling which led to the inclusion of participants with a range of characteristics including those with mild, moderate, and severe PD who have been living with PD over a wide range of years. Whilst the inclusion of participants living with DBS treatment and PD dementia, spousal and non-spousal caregivers who were both

cohabiting or lived separately from the PwP is a further strength of this study, the small number of participants representing each characteristic mean that not all experiences of treatment burden may have been captured. However, there were several limitations. Firstly, this study was conducted in the UK with a publicly funded national health system and the findings may not apply to PwP and caregivers in other countries with different health systems, although they are likely to experience similar challenges worldwide [14]. Secondly, there was a lack of ethnic diversity among participants which may limit the transferability of the findings, although this aligns with the local population of the study region. Thirdly, data regarding financial capacity or deprivation levels were not collected and these factors may influence the experiences of participants. Although reasons for not participating were not recorded, eligible participants with PD who did not respond to the study invitation were aged 67–87 years old, diagnosed with PD between 1–23 years, living alone or cohabiting, with or without a caregiver, and two PwP who had early cognitive impairment. Whilst these were similar characteristics to participants recruited in this study, participants with high treatment burden or less capacity may not have consented to participate in the interviews due to the limited time constraints in their everyday lives trying to manage their PD. Therefore, there may be other aspects of treatment burden and capacity not reported in the findings.

## Implications and next steps

PwP and their caregivers may experience one or more aspects of treatment burden. Therefore, it is crucial to identify which aspects may be most burdensome to allow targeted person-centered interventions to optimize the treatment burden. Achieving patient-centered care for PwP and caregivers through 'Minimally Disruptive Medicine' by developing and implementing flexible models of healthcare system delivery which comprehensively address patient complexity and optimizes healthcare intervention may be helpful [1, 6]. However, a recent systematic review of quantitative interventional studies in adults with long-term conditions reported that only 11 articles evaluated the impact of medical interventions on patient-reported treatment burden [47]. Reduction in medication dosing frequency or providing medical devices that were easier to use in patients with diabetes, the addition of background medication in patients with cystic fibrosis and offering home phototherapy in patients with psoriasis had positive outcomes on treatment burden. Yet, the review reported that only three studies assessed treatment burden as a primary endpoint, and these were all in patients with diabetes. Therefore, further research is needed to evaluate strategies and interventions at both individual and system levels to reduce treatment burden in PD. Whilst several measures such as the Multimorbidity Treatment Burden Questionnaire, Treatment Burden Questionnaire and the Patient Experience with Treatment and Self-Management have been developed to measure treatment burden in long-term conditions, none of these have been validated in PD [48–50]. Determining the extent of treatment burden in PD may help healthcare professionals and researchers determine the factors associated with high treatment burden to target specific interventions or change for PwP and caregivers who may be at risk of poor health outcomes.

## Conclusions

There are potentially modifiable factors that can be implemented by PwP, caregivers, healthcare professionals and healthcare services that may reduce the treatment burden or enhance capacity of PwP and caregivers. Treatment burden and capacity are closely interlinked in PD. Recognition of this by healthcare professionals and adopting a patient-centered approach could improve the experiences of managing PD for PwP and caregivers. This may lead to better health outcomes for those affected by PD.

## Supporting information

**S1 File. Interview guides.**
(PDF)

## Acknowledgments

We would like to thank the study participants for participating in this study and sharing their invaluable views and experiences of living with PD. We would like to thank Angela Dumbleton at the Academic Geriatric Medicine, University of Southampton for transcribing the interviews. For the purpose of Open Access, the author has applied a Creative Commons Attribution (CC BY) licence to any Author Accepted Manuscript version arising from this submission.

## Author Contributions

**Conceptualization:** Qian Yue Tan, Helen C. Roberts, Simon D. S. Fraser, Khaled Amar, Kinda Ibrahim.

**Data curation:** Qian Yue Tan.

**Formal analysis:** Qian Yue Tan, Kinda Ibrahim.

**Funding acquisition:** Qian Yue Tan, Helen C. Roberts, Simon D. S. Fraser, Khaled Amar, Kinda Ibrahim.

**Investigation:** Qian Yue Tan, Simon D. S. Fraser.

**Methodology:** Qian Yue Tan, Helen C. Roberts, Simon D. S. Fraser, Khaled Amar, Kinda Ibrahim.

**Project administration:** Qian Yue Tan.

**Resources:** Helen C. Roberts, Khaled Amar.

**Supervision:** Helen C. Roberts, Simon D. S. Fraser, Kinda Ibrahim.

**Validation:** Qian Yue Tan, Helen C. Roberts, Simon D. S. Fraser, Kinda Ibrahim.

**Visualization:** Qian Yue Tan.

**Writing – original draft:** Qian Yue Tan, Kinda Ibrahim.

**Writing – review & editing:** Qian Yue Tan, Helen C. Roberts, Simon D. S. Fraser, Khaled Amar, Kinda Ibrahim.

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
