## [Decision Letter · Decision Letter 0]

23 Nov 2022

PONE-D-22-24429What are the modifiable factors of treatment burden and capacity among people with Parkinson’s and their caregivers: A qualitative studyPLOS ONE

Dear Dr. Tan,

Thank you for submitting your manuscript to PLOS ONE. After careful consideration, we feel that it has merit but does not fully meet PLOS ONE’s publication criteria as it currently stands. Therefore, we invite you to submit a revised version of the manuscript that addresses the points raised during the review process.

We look forward to receiving your revised manuscript.

Kind regards,

Khatijah Lim Abdullah, DClinP, MSc., BSc

Academic Editor

PLOS ONE

Journal Requirements:

Reviewers' comments:

Reviewer's Responses to Questions

**Comments to the Author**

1. Is the manuscript technically sound, and do the data support the conclusions?

Reviewer #1: Yes

Reviewer #2: Partly

2. Has the statistical analysis been performed appropriately and rigorously? 

Reviewer #1: N/A

Reviewer #2: N/A

3. Have the authors made all data underlying the findings in their manuscript fully available?

Reviewer #1: Yes

Reviewer #2: Yes

4. Is the manuscript presented in an intelligible fashion and written in standard English?

Reviewer #1: Yes

Reviewer #2: Yes

5. Review Comments to the Author

Reviewer #1: L 1 Title: The disorder should be Parkinson's disease, not just Parkinsons. Is it Parkinson's law, gait, syndrome, plus syndrome. A minor point , but needs review.

L31 Abstract methods: Hoehn and yahn stages need a brief explanation

L48: The term long term condion though validate would be better served by the use of chronic disease, which may explain a disorder with no cure relying on palliative care and medications.

L88: The abbreviation QYT has not been explained, confusing

L96: S1 file: Not explained and no indication as to its location, is it an appendix?

L112: How were the participants anonymized, before data collection and analysis? Not stated

L114: What is K1? Not clear

L115: Was a specific qualitatitive framework employed for the inductive data coding and theme generation? Not clear

L119: What type of mind map? Diagram or figure would be appropriate, at least some form of explanation.

L122: QYT only now explained, however, is the female clinician, medical, nursing, paramedical? Not clear

L1299 Pwp, 8 caregivers, now introduced, not previously explained.

L406: Strenths and limitations, purposive sampling shouls also have been mentioned in the sampling method and clarified as to what method employed.

Otherwise no other issues

Reviewer #2: Overview:

While other authors have described treatment burden and the ability to manage said burden, (termed capacity), in the context of PD, this group reports to be the first exploring this relationship in PWP and from care partners and proposes factors which may be modifiable. 9PWP and 8 partners were interviewed and they identified 4 themes related to treatment burden: 1) appointment and access challenges; 2) information gathering; 3) Managing prescriptions; 4) Personal lifestyle changes. Areas relevant to capacity were numerous and included transportation, geography, health literacy, social support, computer literacy and access, and financial support, personal attributes (positive attitude, sense of humor), and faith. Unique to this work is identifying prescription errors, medication availability, collecting prescriptions, and issues with access to GP as novel factors contributing to treatment burden.

Comments:

Any study that attempts to understand and improve quality in complex neurodegenerative conditions is laudable. It is also a logical extension of their previously cited work. My major comment is to consider re-working the discussion so it is more practical for a local PD specialist or even a clinic manager. As it stands, it reads mostly like a reiteration of the results section. Selected examples follow:

How should one approach frequency of healthcare appointments (line 348)? And how would you use the data gathered to support this since the participants seemed to offer conflicting reports: some wanted more and some wanted fewer visits with their specialists.

What about “…medication reviews to reduce polypharmacy may help reduce the treatment burden experienced” (line 352)? Should this be done with the specialist, the GP, a pharmacist working with the treatment team? What change in workflow can be implemented or what strategies can be employed to empower the PWP or dyad to address this?

What about improving health literacy (line 340)? HOW does one go about doing this, and should it fall on the neurologist?

What about maintaining positivity (line 356)? This is nice to encourage but depression and apathy are well-established features of PD, are probably under-diagnosed and under-treated, and should not be ignored.

While not necessary, the authors might consider putting together another figure/table which summarizes these findings in the way they do for the results.

Furthermore, I believe the strengths are oversold. I appreciate that a wide spectrum of disease is represented, and there is considerable burden on analyzing qualitative data, but having one person with DBS is unlikely to represent the collective attitudes of the PD+DBS population. The authors’ limitations are well-stated. “treatment burden amongst caregivers of people with long-term conditions remains understudied” (line 396) was an unfortunate missed opportunity. I do wonder if you would be able to look at zip code of the participants as a proxy for financial status? This may partially address one of the limitations mentioned.

The use of telemedicine in PD specifically has been done, and I point the authors to: Beck, C.A.; Beran, D.B.; Biglan, K.M.; Boyd, C.M.; Dorsey, E.R.; Schmidt, P.N.; Simone, R.; Willis, A.W.; Galifianakis, N.B.; Katz, M., et al. National randomized controlled trial of virtual house calls for Parkinson disease. Neurology 2017, 89, 1152-1161, doi:10.1212/wnl.0000000000004357.

And a trivial comment. Line 69: should read ‘…neurosurgical procedures SUCH AS deep brain stimulation’

6. PLOS authors have the option to publish the peer review history of their article (what does this mean?). If published, this will include your full peer review and any attached files.

Reviewer #1: **Yes: **Manfred Mortell

Reviewer #2: No

---

## [Author Response · Author response to Decision Letter 0]

5 Feb 2023

Dear Academic Editor, thank you for you comment. We have ensured the manuscript meets the PLos ONE style requirements. 

Dear Reviewer #1: Thank you for your comments. Please see our response to the comments below and in the attached 'Rebuttal Letter'. 

- For further clarification we have amended the title as suggested and ensured the term Parkinson’s disease is used throughout the article. 

- We have included a brief explanation for Hoehn and Yahr to indicate the severity of Parkinson’s disease

- The terms ‘long-term condition’ and ‘chronic disease’ may be used interchangeably to depict a health condition with no cure. As such, we have decided to continue with the term long-term condition in this article. 

- We have included a brief explanation that QYT indicates the 1st author 

- The S1 File has been uploaded onto PLOS ONE as per the editorial manuscript requirements for review. 

- As seen in L114, data were anonymised prior to data analysis. 

- We have added an explanation highlighting KI as the last author.

- Data analysis was conducted using thematic analysis, with Braun and Clarke 2006 cited accordingly. This method analysis does not employ the use of a qualitative framework. Rather, familiarisation of data and generation of initial codes systematically and identification of themes are conducted and then re-reviewed to develop final refinement of each theme and relationship between themes. 

- We have briefly explained the use of mindmaps further.

- We have clarified the role of QYT.

- We have now added this information (9 PwP, 8 Caregivers) into the earlier section ‘Participant Recruitment and Sampling’.

- Purposive sampling was described in section ‘Participant Recruitment and Sampling’, page 4, L85 

Dear Reviewer #1: Thank you for your comments. Please see our response to the comments below and in the attached 'Rebuttal Letter'. 

- Thank you for your positive comments and highlighting the novel findings of treatment burden and capacity in Parkinson’s disease that this research study has identified. 

- Your comments regarding the discussion section have been helpful in shaping the revised article. We have rewritten the second and third paragraph of the Discussion section in order to highlight the potential practical examples for change that may be implemented from the study findings. Thank you for the suggestion for an additional figure or table. We gave this serious consideration and have added more to the discussion narrative in the article. 

- We have noted the small sample size for patients with DBS and PD dementia and included this as a limitation. 

- We agree that whilst this work has tried to fill the gap of knowledge around treatment burden amongst caregivers in PD, further research specifically focused on the caregiver treatment burden is required and has been recognised in a review by Sheehan et al (doi: 10.1186/s12877-019-1222-z). 

- In the UK, post codes may be used as a proxy for deprivation status and average disposable income although it is a reflective of the neighbourhood rather than individual circumstances. We have therefore decided not to use this during data analysis. 

- We have included this as a reference and amended the sentence accordingly.

- We have amended the sentence as recommended.

---

## [Editor Report · Decision Letter 1]

15 Mar 2023

What are the modifiable factors of treatment burden and capacity among people with Parkinson’s disease and their caregivers: A qualitative study

PONE-D-22-24429R1

Dear Dr. Tan,

We’re pleased to inform you that your manuscript has been judged scientifically suitable for publication and will be formally accepted for publication once it meets all outstanding technical requirements.

Kind regards,

Khatijah Lim Abdullah, DClinP, MSc., BSc

Academic Editor

PLOS ONE
---

## [Editor Report · Acceptance letter]

23 Mar 2023

PONE-D-22-24429R1 

What are the modifiable factors of treatment burden and capacity among people with Parkinson’s disease and their caregivers: A qualitative study 

Dear Dr. Tan:

I'm pleased to inform you that your manuscript has been deemed suitable for publication in PLOS ONE. Congratulations! Your manuscript is now with our production department. 

Kind regards, 

on behalf of

Dr. Khatijah Lim Abdullah 

Academic Editor

PLOS ONE